# Vision Language Models Cannot Reason About Physical Transformation

## Abstract

Understanding physical transformations is fundamental for reasoning in dynamic, real-world environments. While Vision Language Models (VLMs) show promises in embodied applications grounded in the physical world, whether they genuinely understand physical transformations remains unclear. To address this gap, we introduce *ConservationBench* to evaluate *conservation*—whether physical quantities remain invariant under transformations despite appearance changes. Spanning four quantitative properties (number, length, volume, size), each task requires integrating visual evidence across time and includes counterfactuals where the targeted quantities are not conserved, forming paired conserving and non-conserving scenarios. With systematic variation in prompts, frame sampling methods, and task design, we generate 13,824 questions evaluating on 34 VLMs. Results reveal consistent failure: none demonstrates systematic conservation. Performance remains marginally above chance, with improvements on conservation tasks often accompanied by severe performance on counterfactual controls. This suggests a dependence on superficial patterns or shortcuts over genuine understanding and reasoning on conservation. Moreover, models show no benefit from higher temporal resolution or prompt design. Together, these findings indicate that current VLMs fail to reason about physical transformation.

## 1 Introduction

Recent advances in Vision-Language Models (VLMs) (Zhang et al., 2024b; Radford et al., 2021; Alayrac et al., 2022; Li et al., 2023) have demonstrated remarkable capabilities of perception (Wang et al., 2024b; Chen et al., 2025; Team et al., 2025; Cheng et al., 2024b), reasoning (Zhang et al., 2024a; Xu et al., 2024; Cheng et al., 2024a), and visual commonsense understanding (Zellers et al., 2019; Park et al., 2020). These capabilities hold promise for real-world applications (Brohan et al., 2023), particularly in embodied tasks (Driess et al., 2023; Nasiriany et al., 2024) that demand a genuine understanding of the physical world and its underlying properties (Chow et al., 2025b; Gao et al., 2024). Yet it remains unclear whether VLMs possess a true understanding of physical principles or the capacity to operate reliably in embodied physical environments.

A key factor in human intelligence that enables successful navigation in an embodied, physically grounded world is the ability to understand and reason about physical transformations (Piaget, 1950; 1952; 1965; Baillargeon et al., 1985; 1990; Baillargeon, 1987; 1986; Spelke et al., 1992; Baillargeon & Carey, 2012; Bear et al., 2021; Piloto et al., 2022). This capacity includes tracking objects over time (Spelke et al., 1994; 1995), managing occlusions (Gredebäck & von Hofsten, 2004), and adapting to dynamic environments (Allen et al., 2020). While there are benchmarks evaluating physically plausible video generation (Motamed et al., 2025; Meng et al., 2024a; Yang et al., 2025; Liu et al., 2025; Shi et al., 2024) and physical understanding in VLMs, spanning from everyday scenes (Zheng et al., 2024; Chow et al., 2025a) to high-school physics questions (Wang et al., 2025) and Olympiad-level problems (Qiu et al., 2025; Wang et al., 2025), these efforts focus either on video generation or physical properties in static scenes, leaving underexplored whether VLMs can genuinely reason about physical transformations—where specific properties may or may not remain invariant.

To bridge this gap, we evaluate *conservation* in VLMs—the understanding that physical quantities remain invariant under transformation despite changes in appearance. Here, physical quantity refers to the measurable magnitude of objects along certain dimensions, while spatial transformation de-

Figure 1: Illustrative Tasks and Frame Selection Pipeline in *Conservation Bench*.

notes the continuous process through which objects change in appearance. For example, an agent demonstrating conservation would recognize that pouring water into a differently shaped glass does not alter its volume, despite the change in visible form. Achieving conservation thus requires more than linguistic knowledge of quantity: it demands a systematic understanding that is both reversible and grounded in visual as well as conceptual representations. We introduce *ConservationBench*, a cognitively grounded benchmark for evaluating whether VLMs can reason about physical transformations. The benchmark consists of 192 video-based tasks across four core quantitative properties—number, length, volume, and size—each requiring models to judge whether a quantity is conserved despite visual transformations. To control for shortcut exploitation, we include matched non-conserving controls where the target quantity changes while irrelevant features remain constant. We systematically vary frame extraction method, temporal resolution, and prompting strategy, yielding 36 conditions and 13,824 total trials.

Evaluating 34 VLMs (1B–76B parameters), we find that models consistently fail to integrate temporal information to track conserved properties across dynamic scenes. High accuracy on conservation tasks is often driven by default heuristics, which reverse in non-conserving scenarios, revealing brittle, non-generalizable reasoning. Furthermore, prompting with cues encouraging transformation reasoning or providing higher temporal resolution does not help. These findings expose a fundamental limitation in current VLMs and underscore the need for more grounded, temporally-aware models capable of systematic physical inference.

## 2 RELATED WORKS

### 2.1 EVALUATING AND BENCHMARKING VLMS

The evaluation of vision-language models (VLMs) is central to identifying their limitations and shaping future directions. Early efforts relied on single-task benchmarks such as VQA (Antol et al., 2015), OK-VQA (Marino et al., 2019), MSCOCO (Lin et al., 2014), OCR (Liu et al., 2023), and GQA (Hudson & Manning, 2019). However, with the emergence of multi-modal large language models (VLMs) that claim broader perceptual and reasoning abilities, evaluation has shifted toward holistic benchmarks such as LAMM (Yin et al., 2023), MMMU (Yue et al., 2024), SEED-Bench (Li et al., 2024), and MMBench (Liu et al., 2024b). A growing line of benchmarks focuses specifically on quantity understanding (Rane et al., 2024; Paiss et al., 2023; Rahmanzadehgervi et al., 2024; Yuksekgonul et al., 2022). These tasks typically assess a model's ability to individuate and count discrete objects in static scenes. While useful, such evaluations largely reduce to surface-level enu-

meration. They do not test whether models encode numerical invariance—the understanding that quantity persists across spatial or configurational transformations. In contrast, our work examines whether VLMs go beyond perceptual counting to represent quantity as a conserved relational property. Our work also relates to multi-image and video-based benchmarks (Yue et al., 2024; Song et al., 2024; Jiang et al., 2024; Fu et al., 2024; Liu et al., 2024a; Meng et al., 2024b; Wang et al., 2024a; Huang et al., 2023). These evaluations assess logical reasoning, cross-image comparison, temporal dynamics, and context referencing. While conservation tasks share this multi-image nature, they uniquely target whether VLMs can track continuous physical transformations and recognize the stability of invariant properties across them.

## 2.2 PHYSICAL UNDERSTANDING AND CONSERVATION

Insights from cognitive science underscore conservation as a critical benchmark for systematic physical reasoning. First proposed by Piaget, success on conservation tasks has long been viewed as evidence of emerging mental operations (Piaget & Inhelder, 1969). Developmental studies show that solving these tasks requires constructing transformation-invariant representations while suppressing misleading perceptual cues (Goldin-Meadow & Beilock, 2010; Houdé et al., 2011; Poirel et al., 2012). Behavioral and neurocognitive research further demonstrates that conservation performance depends on sensorimotor grounding and inhibitory control, highlighting the embodied nature of transformation understanding (Beilock & Goldin-Meadow, 2010; Lozada & Carro, 2016). Conservation also builds on more rudimentary abilities such as object permanence and individuation, revealed through studies exploiting the tunnel effect and violation-of-expectation paradigms (Burke, 1952; Flombaum & Scholl, 2006; Noles et al., 2005; Scholl, 2007), which themselves provide essential foundations for robust physical reasoning. In this light, conservation is widely recognized as a foundational cognitive capacity—providing a scaffold for the higher-level physical reasoning needed to navigate dynamic, embodied environments (Fodor, 1975; Baillargeon & Carey, 2012; Barsalou, 2020; Luo et al., 2025).

Recent studies have examined models' abilities to reason about physical properties, causal interactions, and material dynamics (Chow et al., 2025a; Patel et al., 2022; Zheng et al., 2024). Growing evidence suggests that VLMs struggle with fundamental aspects of visual reasoning and physical understanding (Schulze Buschoff et al., 2025; Campbell et al., 2024; Buschoff et al., 2025), with some work exploring how modular frameworks or synthetic training data might address these limitations (Balazadeh et al., 2024; 2025). However, these efforts largely emphasize outcome prediction or descriptive inference, without testing whether models recognize that certain properties remain invariant under transformation. In many cases, success appears to stem from outcome-based heuristics rather than structured mental operations (Newman et al., 2024; Isola et al., 2015). Consequently, it remains unclear whether current VLMs can genuinely integrate sequential evidence to track physical transformations while maintaining stable representations of underlying properties—a core cognitive capacity directly targeted by conservation tasks (Mitchell & Krakauer, 2023).

## 3 EXPERIMENTAL DESIGN

### 3.1 CONSERVATION TASKS

To systematically measure the conservation ability of VLMs–the understanding that specific physical properties remain invariant under transformations despite changes in appearance, we construct a suite of conservation tasks in the form of videos that visually depict physical transformations across four fundamental quantitative properties.

- **Conservation in Number**: Two rows of identical coins are presented in an initial configuration. One row is then spread apart without adding or removing any coins. The task assesses whether the quantity is perceived as unchanged despite the altered spacing.
- **Conservation in Length**: Two straws of identical length are shown in an initial configuration. One of the straws is then repositioned without altering its actual length.
- **Conservation in Volume**: A fixed volume of liquid is poured from one container into another of a different shape. Although the height changes significantly, the volume remains constant throughout the transformation.

- **Conservation in Size**: A lump of playdough is reshaped from one form (e.g., a ball) into another (e.g., a flattened disc). While the shape and surface features change, the total mass remains the same across both states.

Although the four conservation types probe distinct physical properties, the tasks follow a unified structure: a transition from an initial to a final state mediated by an observable transformation. Each video begins with an initial state, proceeds through a continuous transformation (e.g., pouring, spreading, flattening), and ends with a new state where the surface appearance of the object of interest is altered. This design mirrors real-world scenarios where physical reasoning depends on integrating perceptual evidence across time.

**Generalization across Task-irrelevant Features**   To ensure the robustness and generalizability of the conclusions drawn from our benchmark, we systematically vary key visual parameters in each conservation task (Table 3). These parameters include object count, size, color, layout, container shape, and the direction of transformation. Each conservation property consists of 48 unique video instances of different configurations, resulting in a total of 192 videos. This controlled variation guarantees that the core conservation principle is preserved across a wide range of visual contexts, thus preventing models from relying on memorized templates or superficial cues.

**Transformation-mandatory and Transformation-helpful**   Notably, conservation tasks differ in how strongly they depend on observing the transformation. We classify them into two categories: *transformation-mandatory* and *transformation-helpful*. In mandatory tasks (volume and size), witnessing the transformation is essential—for instance, in volume conservation, seeing the liquid poured is necessary, since the final height alone is insufficient for judging quantity. In helpful tasks (number and length), correct judgments can still be made from the initial and final states, as the relevant quantity remains visually accessible despite superficial changes. This distinction enables a more diagnostic evaluation: models that excel on helpful but not on mandatory tasks may rely on static cues rather than forming internal representations of the transformation process.

To this end, we further curated a set of 96 tasks derived solely from the final frame of transformation-helpful tasks. Here, models are prompted to compare numbers and lengths directly based on simple counting and intuitive judgments of spatial extent. This design isolates pre-conceptual, rudimentary forms of quantitative assessment—such as item enumeration and perceptual matching—from the broader representational demands of transformation-based reasoning. By contrasting performance on these static tasks with temporal conservation trials, we can reveal how basic quantitative sensitivity relates to the more systematic representations of quantity that underlie conservation reasoning.

### 3.2 Non-conserving Tasks

A key limitation of applying conservation tasks to model evaluation is the uniformity of ground-truth labels: since all standard tasks involve quantity preservation, models can appear accurate simply by defaulting their responses to indicate invariance, due to biases from either visual contexts or linguistic patterns in the prompts, without genuinely reasoning about the physical transformation itself (Li et al., 2025b). To address this, we create non-conserving counterfactuals as a set of controlled experiments where the quantity of interest is explicitly altered during the transformation without changing the task-irrelevant features. That is to say, these manipulations are performed within the same environments, using identical object sets and visual contexts, thereby ensuring a controlled comparison. This design enables fine-grained assessment of model sensitivity to actual changes in quantity, rather than reliance on superficial heuristics or distributional priors. We describe the construction of these control tasks across each property below.

- **Non-conserving Transformation of Number**: Two rows of identical coins are presented in an initial configuration. One row is then spread apart, with one coin added to said row.
- **Non-conserving Transformation of Length**: Two straws of identical length are shown in an initial configuration. One of the straws is then repositioned while its actual length is altered (extendable straws are used).
- **Non-conserving Transformation of Volume**: A fixed volume of liquid is poured from one container into another of a different shape. A significant portion of water is left in the original container instead of being completely poured in.

- **Non-conserving Transformation of Size**: A lump of playdough is reshaped from one form (e.g., a ball) into another (e.g., a flattened disc). A significant portion of playdough is left in the experimenter's hand without being integrated into the new shape.

Following this design, we curated a control set in which each non-conserving control is paired with a conservation task under matched configurations, yielding another 192 videos.

Table 1: **Overview of Multi-image Task Conditions and Evaluation Scale**

| **Core Dataset** | 384 total videos |
|---|---|
| Conservation Tasks | 48 videos × 4 properties = 192 videos |
| Non-conserving Control | 48 videos × 4 properties = 192 videos |
| **Multi-frame Curation** | 36 conditions (factorially combined) |
| Extraction Method | 3 Conditions (Uniform, SEVILA, Human) |
| Frame Count | 3 Conditions (3 frames, 8 frames, 16 frames) |
| Prompting | 4 Conditions (Direct Question, "Sequential", CoT, "Continuous") |
| **Total Task Trials** | $384 \times 36 = 13,824$ evaluation trials |

### 3.3    ADAPTATION TO MULTI-FRAME INPUT

#### 3.3.1    TEMPORAL RESOLUTION

The ability to understand physical transformations critically depends on comprehending dynamic processes over time. Unlike static snapshot reasoning, robust comprehension requires recognizing continuity across successive observations. Human perception benefits from high frame rates (e.g. $\sim$ 30-60 frames per second) that convey rich temporal information, while the architectural and computational limitations of VLMs restrict them to inferring such dynamics from discrete and often sparse inputs.

To investigate the impact of temporal resolution on conservation understanding, we vary the number of frames extracted from each video:

- **3-frame condition**: Only three frames are provided—the first, the last, and one intermediate frame. This condition presents minimal temporal information while retaining just enough cues for humans to solve the task.
- **8-frame condition**: Eight frames are sampled to offer moderate temporal granularity. This condition is designed to contrast qualitatively with the 3-frame condition by enabling multi-frame representations of the temporally continuous scene.
- **16-frame condition**: Sixteen frames are sampled to provide finer-grained temporal information, offering a more detailed depiction of the transformation process, contrasting quantitatively with the 8-frame condition.

In all conditions, the first and last frames are fixed to preserve the initial and final states of the transformation. This manipulation enables us to assess whether increased intermediate visual evidence regarding the transformation process enhances the model's ability to infer conservation.

#### 3.3.2    SAMPLING STRATEGY

In studying physical transformations, the sequence and selection of visual inputs are crucial. Due to computational limitations, state-of-the-art VLMs are optimized for sparse, multi-frame reasoning. This raises an important question: do different frame selection strategies influence the model's understanding of dynamic scenes? Additionally, do humans and models rely on different criteria when identifying informative visual moments? To explore these questions, we implement and compare three frame extraction strategies, each reflecting distinct assumptions about what defines a "representative" moment in a physical event.

- **Uniform Sampling**: Frames are sampled uniformly across the timeline, serving as a baseline approach commonly used in prior work, based on the assumption that temporal regularity sufficiently represents informational diversity.

- **Human-based**: To obtain a baseline for human intuition in frame extraction, we recruited N = 18 annotators. Each annotator was randomly assigned a subset of the dataset and asked to manually select the intermediate frames that captured the essential stages of the transformation.
- **Model-based**: We adopt SEVILA (Yu et al., 2023) and leverage a BLIP-2-based Localizer to identify language-aware keyframes. Prompted with the same instruction assigned to humans ("extract the most complete set of frames that capture the entire process"), the Localizer module selects frames with high relevance scores, which are then passed to the Answerer module for inference. This method formalizes a strategy akin to semantic salience: choosing frames that are maximally informative given a specific query.

This design allows us to test whether different frame selection strategies affect model performance on physical transformation reasoning. We hypothesize that optimizing frame selection, rather than merely increasing frame quantity, leads to more effective representations of dynamic events. We detail our data curation process in Appendix A and prompting strategies in Appendix B, redand provide example input in Appendix C.

## 4 EXPERIMENTS

### 4.1 INFERENCE AND EVALUATION

**Inference.** We evaluate 34 VLMs spanning diverse model architectures, training data, and parameter scales, covering both mainstream commercial systems and advanced open-source models. To ensure fidelity, comparability, and reproducibility, we strictly adhere to reference configurations and implementations from the official codebases. Refer to Appendix D for further details.

**Evaluation.** To evaluate free-form outputs of VLMs on multiple-choice questions (MCQs), we follow the two-stage scoring method of Li et al. (2025a). In Stage 1, each VLM output is mapped to a unique choice from the provided options or labeled FAIL when no unambiguous mapping is possible. Mapping follows a hybrid strategy: deterministic template matching is applied first, and unresolved cases are adjudicated by an LLM-as-a-Judge constrained to the option set. Models exhibiting persistently high FAIL rates are excluded from further analyses to avoid bias from nonsensical outputs. In Stage 2, the mapped option is compared against the ground-truth answer, with all FAILs scored as incorrect. Details are provided in Appendix E.

### 4.2 HUMAN BASELINE

Given the large number of questions and the cost of human annotation, we curated a representative subset by randomly selecting one out of every eight task configurations for each quantitative property, counterbalanced across conservation tasks and non-conserving controls. This resulted in a total of 864 questions. We hypothesize that reduced variation in task-irrelevant features is unlikely to compromise the benchmark's validity or generalizability given the robustness of human reasoning. Participants received the same stimuli and three-choice questions as the VLMs, with the exception that they directly selected answers rather than requiring LLM judge parsing. The aggregated human accuracy reaches 95.25%, consistent with decades of developmental research showing that humans from late childhood reliably solve conservation tasks with near-perfect accuracy (Piaget, 1965; Houdé, 1997; Pezzulo et al., 2013; Viarouge et al., 2019). These results validate our benchmark design and its adaptation for evaluating VLMs.

### 4.3 MAIN RESULTS

As shown in Figure 2A, model accuracy across 34 VLMs ranges from 27% to 49%, with most performing only marginally above the 33.3% chance level. In contrast, human participants exceed 95% accuracy (Section 4.2), highlighting a clear gap between VLMs and intuitive human reasoning. Even top-performing models (e.g., INTERNVL-2-8B) fail to generalize across conservation and non-conserving controls. These limitations generalize across all four quantitative domains—number, length, volume, and size (see Appendix G for details). While number and length yield marginally better results, no model demonstrates consistent success across domains. Collectively, these results reveal a core limitation: VLMs struggle to integrate temporal cues or track invariant properties through dynamic transformations, a key requirement for grounded physical reasoning.

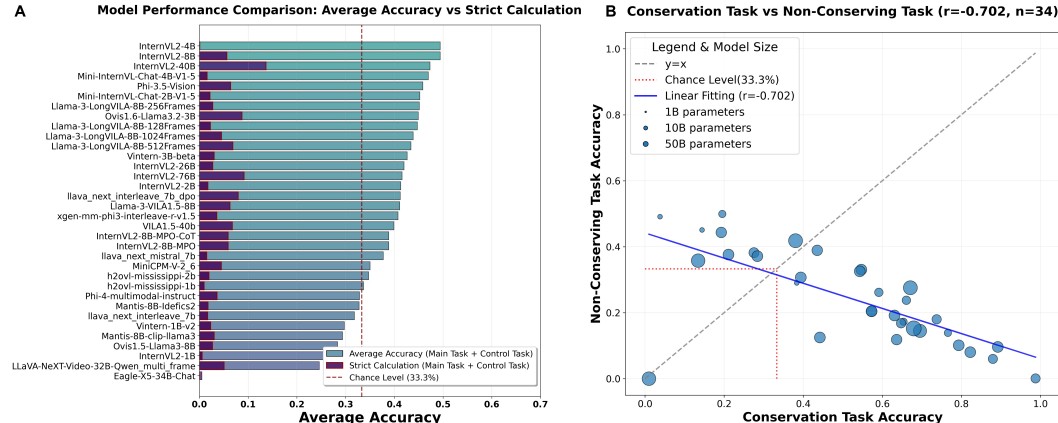

Figure 2: Overall Performance on *ConservationBench*. A. Accuracy averaged across conservation tasks and non-conserving control compared to strict pairwise calculation; B. Performance on non-conserving control in relation to conservation tasks.

### 4.3.1 INVERSE PERFORMANCE ACROSS CONSERVATION AND NON-CONSERVING CONTROL

By comparing model performance on non-conserving control tasks against conservation tasks, we observe a strong negative correlation: models that perform better on conservation tasks tend to perform worse on the corresponding control tasks, and vice versa. This trade-off indicates a systematic bias toward different interpretations of the task scenario—models tend to "default" almost randomly across the axis of conserving vs. non-conserving heuristics, regardless of actual transformation evidence (Figure 2B). Crucially, this pattern reveals a diagnostic failure: models are not simply underperforming but are exhibiting mutually offsetting errors across matched task types.

We further validate this pattern using a strict pairwise evaluation across the full set of 6,912 matched conservation and non-conserving control tasks (as shown in Figure 2A; labeled in purple). In this analysis, a model is only marked correct if it answers both tasks in a pair correctly—capturing whether it can jointly recognize quantity preservation and detect meaningful violations under matched visual conditions. We found that all models perform well below chance, uniformly achieving accuracy rates under 10%, indicating that they are unable to reliably distinguish between conserving and non-conserving scenarios. For example, the second best-performing model on the standard benchmark (INTERN-VL2 4B), which reached nearly 50% average accuracy, drops to 0.001% in this joint evaluation, suggesting that its success on conservation tasks is driven almost entirely by a strong bias over quantity invariance rather than genuine reasoning about physical transformations. This finding further supports the conclusion that models fail to internalize structured physical reasoning and instead rely on brittle default strategies of quantity assessment.

### 4.3.2 DISSOCIATING SOURCES OF BIAS.

We show that model performance on conservation tasks is largely driven by shallow heuristics rather than grounded physical reasoning. To disentangle the source of this bias—whether it arises from visual features or linguistic priors—we rerun the same prompts using fully white, content-free images, while keeping all textual inputs constant. Model responses were evaluated as if they were answering standard conservation tasks. If performance were driven by visual cues, models should operate at chance when visual input is removed. Conversely, systematic deviations from chance would indicate a reliance on linguistic biases embedded during pretraining—favoring either conservation (bias toward invariance) or non-conservation (bias toward perceptual change).

The results reveal heterogeneous linguistic biases across models, with no clear relationship to overall performance on the original benchmark (see Appendix H for details). Most models exhibit strong priors, systematically favoring either conservation or non-conservation even in the absence of any visual content, while a smaller subset performs near chance, suggesting greater reliance on visual

features. These divergent patterns underscore the influence of pretraining data and architectural inductive biases, independent of any genuine capacity for physical reasoning.

### 4.3.3 DOES SCALING OF MODEL SIZE HELP?

The advancement of LLMs has been closely tied to the empirical scaling law—predictable power-law improvements in performance with increased compute, parameters, and training data (Kaplan et al., 2020; Henighan et al., 2020; Zhai et al., 2022)—as well as emergence, the abrupt appearance of qualitatively new abilities as models grow larger (Wei et al., 2022; Aghajanyan et al., 2023; Bubeck et al., 2023; Berti et al., 2025). This raises a natural question: *Does the capacity to understand physical transformations and conservation similarly emerge with scale?* Our results suggest not: performance shows no significant relationship to model size, with larger models exhibiting substantial variability in accuracy across both conservation tasks and non-conserving controls (as shown in Figure 2B). These findings indicate that scaling alone is insufficient for current VLMs to acquire the capacity for genuine reasoning about physical transformations.

### 4.4 DIFFERENT PROMPTING STRATEGIES, FRAME NUMBERS, AND SAMPLING METHODS

We further analyzed model performance across three experimental factors—prompt type, frame count, and frame sampling method—evaluated separately for Number & Length versus Volume & Size conservation tasks. We highlight the main conclusions as below.

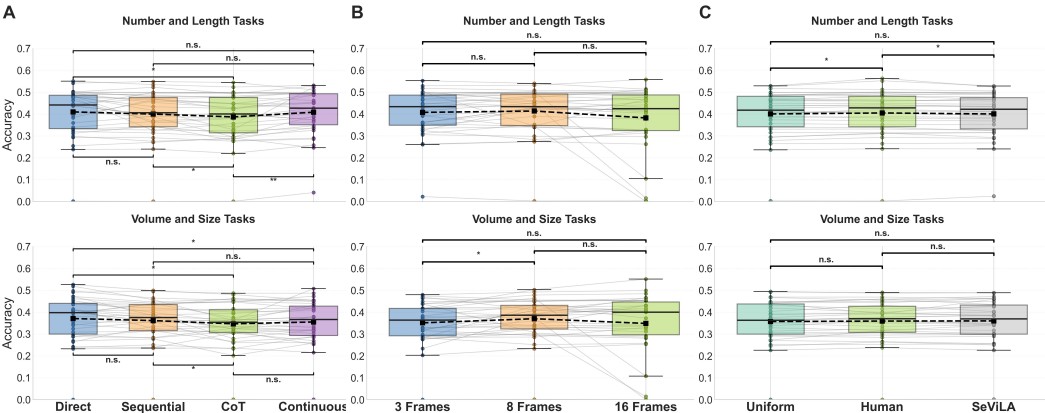

Figure 3: Model performance showing main effects by (A) prompt type, (B) number of frames, and (C) frame sampling method. Each panel averages across the other two factors from the full factorial design (4 prompts × 3 frame counts × 3 extraction methods).

**"Continuous" cue fails; CoT makes it worse.** We evaluate the impact of different prompting strategies across both conservation tasks and non-conserving controls (Figure 3A). Averaged across quantitative properties, linguistic scaffolding provides limited benefit: none of the three prompting types—Sequential, Chain-of-Thought (CoT), or Continuous—outperform the Direct Question baseline. In fact, performance is significantly worse when using conceptual cues that frame transformations as continuous processes. For Number and Length tasks, Sequential prompts lead to significantly lower accuracy than Direct questions ($t = 2.947$, $p = 0.00585$), as do CoT ($t = 2.701$, $p = 0.01083$) and even Direct vs. Continuous ($t = 2.663$, $p = 0.01188$), all indicating reliable performance drops. CoT prompting performs the worst overall, suggesting that explicitly verbalizing the reasoning process may amplify reliance on brittle heuristics. These results indicate that prompting alone does not improve—and may even impair—transformation reasoning in VLMs.

**Increased Temporal Resolution Helps Little.** We assessed whether increasing temporal resolution improves model performance by comparing tasks rendered with 3, 8, or 16 visual frames (Figure 3B). Across Number & Length tasks, no statistically significant improvements were observed (all $p > 0.05$), and only a marginal effect was found for Volume & Size: models performed slightly better when given 8 frames compared to 3 frames ($t = -2.075$, $p = 0.04583$). This suggests that

simply providing more visual information over time in general does not enhance models' capacity to track or reason about physical transformations. Despite higher frame counts offering richer depictions of dynamic processes, current VLMs appear unable to integrate temporally extended input. Instead, they continue to rely on static visual heuristics extracted from individual frames, regardless of sequence length or transformation complexity.

**Human-aligned Frame Extraction Aids Performance.** Across the Number & Length tasks, models achieve significantly higher accuracy when presented with frames selected by humans based on intuitive understanding of the transformation, compared to both uniform temporal sampling ($t = -2.526$, $p = 0.01653$) and LLM-based extraction using SEVILA ($t = 2.523$, $p = 0.01664$). However, this advantage did not extend to the Volume & Size tasks, where no significant performance differences were observed across frame selection strategies (Figure 3C). These findings suggest that human-aligned frame extraction can aid performance in tasks where transformation cues are helpful but not essential, likely by highlighting snapshots that facilitate heuristic-based reasoning or static quantity assessment. In contrast, when success requires tracking continuous transformations, such selection strategies confer no benefit.

### 4.5 INSIGHTS FROM TRANSFORMATION-HELPFUL TASKS

Number & Length tasks, categorized as Transformation-helpful, may not necessarily require understanding the physical transformation process, as judging the final frame alone can suffice. For example, the Number task can be solved by directly counting the coins in the last frame. Thus, it is important to evaluate whether conservation plays a significant role in Transformation-helpful tasks or whether they can be solved by simply counting or perceiving length in the last frame. To explore this, we compare model performance on multi-frame versus last-frame-only inputs.

We found no significant difference in mean accuracy between the two conditions across models (both $p > 0.05$). However, the last-frame-only condition exhibited substantially greater variance (Number: $Std = 0.0750$ vs. $0.1861$; Length: $Std = 0.0810$ vs. $0.1490$), with some models performing notably better than others (see further details in Appendix I). This suggests that certain models may be more attuned to static quantity assessment (e.g., counting or length comparison), effectively solving tasks using cues from the final frame alone. Yet, these same models perform markedly worse on multi-frame tasks involving continuous physical transformations—even when such reasoning is not strictly necessary. This underscores that effective reasoning about physical processes requires more than simple heuristics. In fact, the inability to engage in such reasoning may nullify any advantage in static assessments when tasks demand integration of temporal dynamics.

## 5 CONCLUSION

This study introduces a cognitively inspired benchmark that systematically evaluates whether VLMs can reason about physical transformations through both conservation tasks and non-conserving controls. Our findings reveal that current models are consistently incapable of integrating temporal information to track physical properties across dynamic visual inputs. Models achieving high performance on conservation tasks do so by relying on default heuristics over quantitative assessment, leading to inverse performance trade-offs when confronted with non-conserving scenarios. These results underscore a fundamental gap in structured physical reasoning and point to a key challenge for developing more grounded, temporally aware AI systems capable of systematic inference in real-world environments.

Conservation tasks offer a foundational benchmark for evaluating whether models can reason about physical transformations in quantitative domains. Future work may extend this paradigm to more complex physical settings, including richer object dynamics, multimodal inputs, or uncertainty, and evaluate conservation reasoning in goal-directed contexts such as planning or tool use. Such extensions are crucial for testing whether VLMs can support structured physical reasoning in real-world scenarios.

## 6 REPRODUCIBILITY STATEMENT

**Data Availability and Transparency:** We provide complete documentation of our data generation process, including precise descriptions of task construction, sampling procedures, and quality control protocols. Representative samples for each quantitative domain are included in the supplementary material. The full dataset—comprising all 13,824 tasks spanning conserving and non-conserving variants—will be released publicly upon publication to ensure transparency and enable direct replication of our experiments.

**Evaluation Pipeline Standardization:** All evaluations are conducted using publicly released VLMs with default settings from their official implementations. Our pipeline adheres to established evaluation standards and reproduces published results on existing benchmarks such as MMBench using the same configurations. This alignment validates the reliability of our evaluation setup and ensures fair comparisons across all models in our benchmark.

## 7 ETHICS STATEMENT

This research presents no identifiable ethical concerns. It involves purely computational evaluations of publicly available VLMs using synthetic image-text inputs. No human subjects, private data, or sensitive content are involved at any stage. All models evaluated are publicly released, and all tasks are generated programmatically under controlled conditions to avoid harm or misuse.

## 8 LLM USAGE STATEMENT

Large Language Models (LLMs) were used to assist with the writing and editing of this manuscript. Specifically, an LLM was employed to refine language, improve clarity, and enhance the overall readability of the text. Tasks included grammar correction, sentence rephrasing, and improving narrative flow across sections. Importantly, the LLM was not involved in any part of the research process, including ideation, methodological design, or data analysis. All scientific content, experimental decisions, and conceptual contributions were developed solely by the authors. The authors retain full responsibility for the content of the manuscript, including any text edited or suggested by the LLM. We confirm that the use of the model adheres to ethical guidelines and does not constitute plagiarism or scientific misconduct.

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

# APPENDIX

## A  DATA CURATION

**Curation and Quality Control.** *ConservationBench* was curated by three annotators with college-level training in cognitive science or computer science. Each video underwent two independent cross-review passes; items failing to meet design criteria were removed or revised.

**Data Acquisition.** All videos were captured under standardized recording conditions using a fixed camera setup, with consistent lighting and background held constant within each property category. Each transformation was carefully scripted to ensure visual clarity, reproducibility, and minimal ambiguity.

**Design Principles.** To ensure conceptual integrity and interdisciplinary rigor, we adopt three design criteria for each item: (i) *Discriminativeness*—tasks are constructed such that models lacking the targeted knowledge are systematically driven toward incorrect responses; (ii) *Minimal confounding*—instances are designed to minimize reliance on ancillary skills (e.g., object recognition); and (iii) *Minimal textual shortcuts*—tasks cannot be solved using textual cues alone and instead require genuine multimodal reasoning.

## B  PROMPTING STRATEGY

Reasoning about conservation often requires interpreting the transformation as a continuous process across the videos or sequence of frames. To examine how prompts influence temporal integration and transformation-based reasoning, we design four prompt types, each progressively enhancing the model's awareness of the underlying continuous process, as summarized in Table 2.

Table 2: **Four different prompt format used in our benchmark.**

| Prompt Type | Prompt Example |
|---|---|
| Direct Question | Is the number of coins in the upper row the same as in the lower row in the final image? |
| "Sequential" Prompt | Please process the images below sequentially, and then answer: Is the number of coins in the upper row the same as in the lower row in the final image? |
| CoT Prompt | Please process the images below sequentially. First describe what happens across the images, then answer: Is the number of coins in the upper row the same as in the lower row in the final image? |
| "Continuous" Prompt | The above images represent a continuous process. Please answer: Is the number of coins in the upper row the same as in the lower row in the final image? |

Together, these prompting strategies enable us to evaluate how different forms of linguistic scaffolding shape model engagement with visual dynamics. The "Sequential" and CoT prompts encourage frame-by-frame perception with step-by-step reasoning, directing attention to frame-wise visual evidence. In contrast, the "Continuous" prompt explicitly presents the multi-frame input as a continuous process, offering a conceptual cue to support conservation reasoning.

## C  EXAMPLE INPUT

To provide clarity on the exact format of inputs provided to models, we present a complete example task below, including both the visual frames and the full textual prompt.

**Task:** Conservation of Number (Conserving condition)

**Task Configuration:** This example demonstrates a Number conservation task using Uniform extraction method with 8 frames and Direct Question prompt format.

**Visual Input:** The model receives a sequence of frames extracted from the video in temporal order (Frame 1 through Frame 8), ensuring that the transformation process is presented chronologically without any frame order disruption. Figure 4 shows an example with 8 frames, where frames are sampled uniformly across the video timeline. The first frame shows the initial state (two rows of coins with equal numbers), intermediate frames capture the transformation process (spreading one row), and the final frame shows the end state (one row spread out while maintaining the same number of coins).

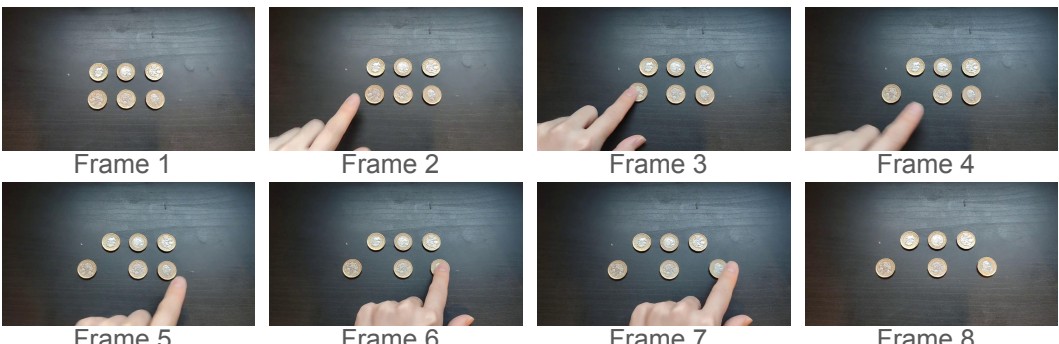

Figure 4: **Example visual input:** A sequence of 8 frames from a number conservation task, showing the initial state, transformation process, and final state.

**Textual Input:** Below is the structure of the prompt provided to the model (using the "Direct Question" format). The [Image] placeholders indicate where the corresponding frames from Figure 4 are embedded in the actual input:

```
Frame 1: [Image]
Frame 2: [Image]
Frame 3: [Image]
Frame 4: [Image]
Frame 5: [Image]
Frame 6: [Image]
Frame 7: [Image]
Frame 8: [Image]

Is the number of coins in the upper row the same as in the lower row
in the final image?

Please choose one of the following options:
(A) No, the lower row has more coins.
(B) No, the upper row has more coins.
(C) Yes, they are the same.
```

**Ground Truth:** Option (C) - Yes, they are the same.

**Alternative Prompt Formats:** For other prompt types, the question is prefixed with additional instructions. For example, the "Sequential" format would begin with "Please process the images below sequentially, and then answer: [question]", while the CoT format would include "Please process the images below sequentially. First describe what happens across the images, then answer: [question]". See Table 2 for details on all four prompt formats.

## D    MODEL INFERENCE

We evaluate 34 VLMs spanning diverse architectures, training regimes, and parameter scales, including mainstream commercial systems and advanced open-source models ranging from 1B to 76B parameters. Inference is conducted on a cluster equipped with 8× NVIDIA H100 (80 GB) GPUs. As a practical policy, models of 1–13B parameters typically run on a single GPU; 13–32B on two GPUs; 32–70B on four GPUs; and >70B on all eight GPUs.

To preserve fidelity and reproducibility, we adhere to configurations and reference implementations from the official codebases, avoiding unnecessary modifications. We build a scalable evaluation framework supporting parallel execution and compartmentalized environments. Inference jobs are distributed across GPUs via a dynamic scheduler that maximizes utilization and minimizes idle time. We additionally develop a lightweight modality-verification suite that prompts each model to summarize the media information it receives, and then the responses are checked by human reviewers to verify correct input routing and modality handling in our inference pipelines.

## E    EVALUATION

Rule-based matching degrades when model outputs are complex (e.g., chain-of-thought), yielding elevated false positives/negatives and requiring continual template optimization to cover corner cases. LLM-based matching better identifies the intended choice within complex free-form text, but it can hallucinate—especially when a brief answer is embedded within extensive context. To balance these trade-offs, we introduce Hybrid Matching, which prioritizes deterministic template matching to extract answers from VLM responses and, on failure, falls back to an ensemble of four LLM judges (Qwen2.5-72B-Instruct, Mixtral-8x7B-Instruct-v0.1, DeepSeek-R1-Distill-Llama-70B, and Llama-3.1-70B). The ensemble decision is accepted only if at least three of four models return a consistent extraction; otherwise, the mapping is deemed unsuccessful. By coupling the precision of regular-format extraction with the semantic flexibility of LLM adjudication, Hybrid Matching delivers more robust and reliable mappings across diverse response styles.

# F COUNTERBALANCING CONDITIONS

Table 3: **Counterbalanced variations of task-irrelevant features** Each unique combination of parameter values yields 48 distinct task instances per domain.

| Domain | Parameter | Variations |
|---|---|---|
| Number | P1: Object Type | 2 variants (Uniform, Mixed) |
| | P2: Mapping Shift | 2 variants (Lower vs. Upper row moved) |
| | P3: Distance Spread | 2 variants (Near, Far) |
| | P4: Number of Objects | 6 variants (3–8 coins) |
| | Total combinations: | $2 \times 2 \times 2 \times 6 = 48$ |
| Length | P1: Object Type | 2 variants (Uniform, Mixed) |
| | P2: Mapping Shift | 2 variants (Lower vs. Upper straw moved) |
| | P3: Distance Moved | 2 variants (Near, Far) |
| | P4: Direction | 2 variants (Left, Right) |
| | P5: Transformation Action | 3 variants (Slide, Rotate, Vertical) |
| | Total combinations: | $2 \times 2 \times 2 \times 2 \times 3 = 48$ |
| Volume | P1: Liquid Color | 8 variants |
| | P2: Glass Transaction | 2 variants (Tall $\rightarrow$ Short, Short $\rightarrow$ Tall) |
| | P3: Liquid Volume | 3 variants (Small, Medium, Large) |
| | Total combinations: | $2 \times 8 \times 3 = 48$ |
| Size | P1: Object Color | 8 variants |
| | P2: Shape Transformation | 6 variants (Crossing Sphere, Cylinder, Plane) |
| | Total combinations: | $6 \times 8 = 48$ |

## G COMBINED MODEL PERFORMANCE BY QUANTITATIVE PROPERTIES

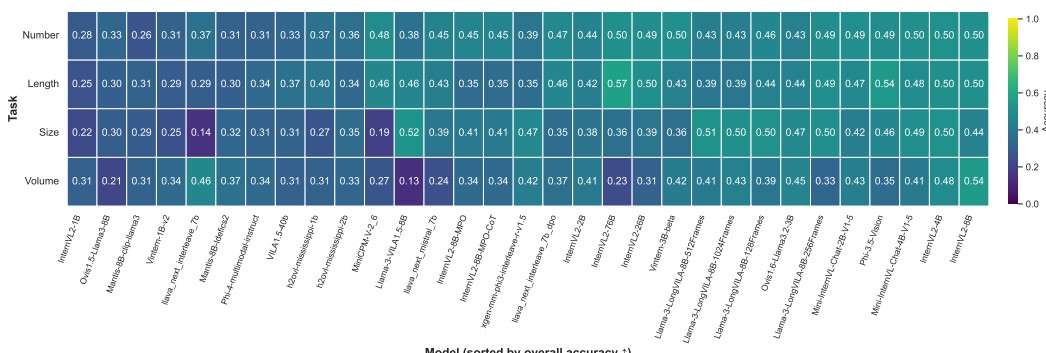

Figure 5: **Model performance by conservation task properties.** Models consistently perform around chance accuracy and significantly below humans, with no significant differences across quantitative properties.

## H MODEL RESPONSE ON WHITE CONTROL AS COMPARED TO CONSERVATION TASK SCENARIOS.

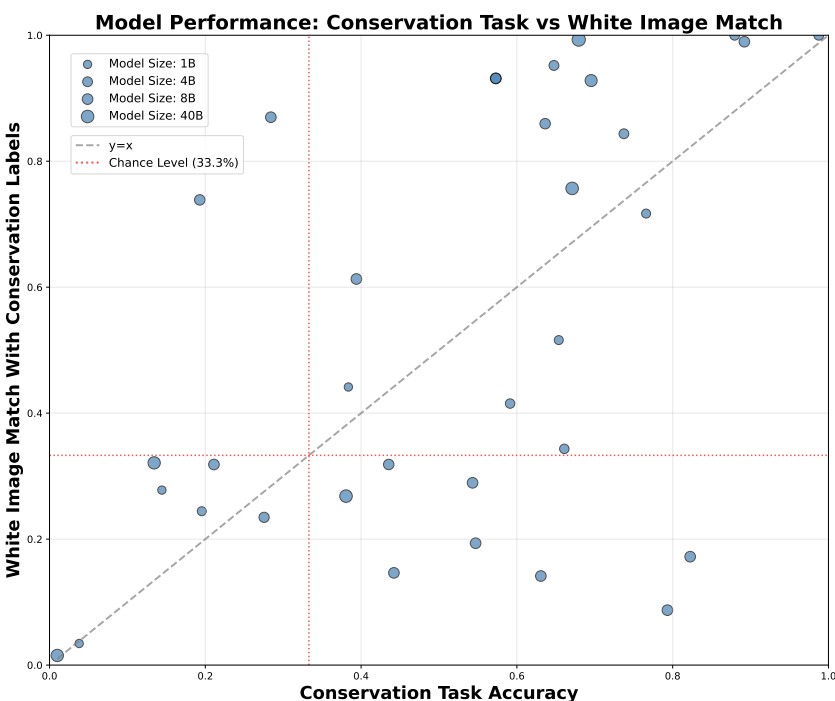

Figure 6: **Model response on white control as compared to conservation task scenarios.** We assess model response to prompts asking about physical transformation, match them in accordance to conservation labels, and compared their accuracy with conservation task performances, revealing distinct sources of bias across linguistic and visual features.

# I COMPARING MULTI-FRAMES VS LAST FRAME PEROFRMANCE ON TRANSFORMATION-HELPFUL TASKS

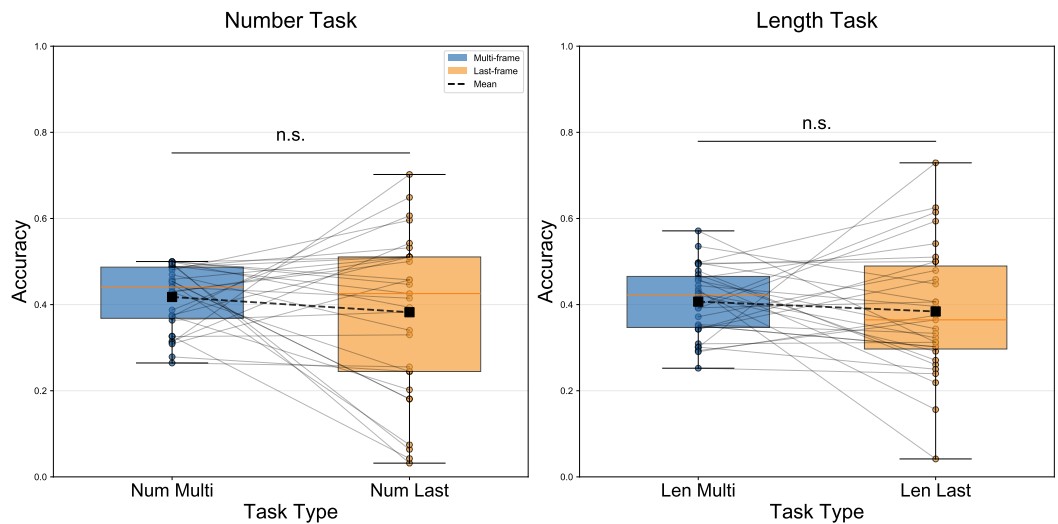

Figure 7: **Model performance on Multi-frames vs Last Frame Conditions in Transformation-helpful Tasks.** While mean accuracy does not differ significantly between the two conditions, variance is substantially higher in the last-frame-only setting.

