# OpenReview forum: "Vision Language Models Cannot Reason About Physical Transformation"
_ICLR.cc/2026/Conference — Submitted to ICLR 2026_

### Official Review · Reviewer_1A1m · 2025-10-26

**Soundness:** 3
**Presentation:** 4
**Contribution:** 3
**Rating:** 6
**Confidence:** 5

**Summary:**

This paper explores whether vision-language models (VLMs) can reason about physical conservation - that the quantitative properties of objects, such as their number, length, volume, and size remain invariant over short time periods. A large benchmark of videos and extracted frame sequences is constructed, and used to study 34 VLMs. They find that no VLMs can robustly reason about conservation, at least not to the level of the average human. This is a surprising result given the established success of VLMs on large-scale general benchmarks and the triviality (for humans) of the conservation tasks being used.

**Strengths:**

This paper is methodologically sound, and is an incremental contribution to a growing literature exploring the physical reasoning capabilities of VLMs. Particular strengths are:
* The rigorous control conditions used throughout to test alternative explanations for VLM model performance.
* The large number of open-source models used.
* The use of a meaningful human baseline for comparison.

**Weaknesses:**

The paper has a number of weaknesses:
1. The hybrid evaluation is an interesting solution to the problem of evaluating complex outputs, but using LLM judges incurs significant overhead for the practitioner. Since the paper currently relies only on open-source models (as far as I can tell) and the benchmark uses multiple choice questions, the authors could simply use the log-probability of the choice label, conditional on the text-image input. This could be normalised across the possible outcomes too. This is quite standard in benchmark evaluation (see EleutherAI's lm-evaluation-harness).
2. Many new models are also capable of processing videos. I would suspect that the next generation of VLM will process video data effectively. Therefore, the longevity of the benchmark would be ensured if video-based evaluation was a central contribution. The authors could conduct a small study with, say, Qwen-2.5-VL-7B (an open source video-language model) to see if there is any difference between performance on video and performance with frame sequences.
3. Many of the tasks used here might be quite novel to the language model, so it's not clear whether these models *could* learn to reason about conservation if given the right training, or whether there is something deeper about architecture/pre-training that prevents them from doing so. Two further experiments are required to elucidate this. First, examining whether in-context learning can boost model performance. Here, examples of conservation/non-conservation with correct labels are given sequentially prior to the test question. Second, examining whether supervised fine-tuning on a subset of the tasks improves performance. There are three conditions of interest here: training on (a) a random subset of the tasks; (b) conservation problems; (c) non-conservation problems. I suspect, however, that fine-tuning would be just as brittle (see Schulze-Buschoff et al. 2025). This would make a more powerful point about VLMs. It's not just that current models off-the-shelf are incapable, but it's a feature of the architecture/large-scaled pre-training that prevents them from having these common-sense intuitions.
4. I don't really think the white-image condition is a good control. To me, the bias towards invariance is shown by the delta between the conservation and non-conservation conditions. I would not expect systematic deviations from chance with white images, because where would that systematicity come from, absent some visual input? Perhaps I am missing the logic here.
5. There is some missing literature, which I include below.
6. It's a semantic point, but I dispute that the main contribution of this paper is a benchmark. Rather, it's a careful series of experiments to test some hypotheses about MLLM capabilities. I don't think anyone will want to use this as a benchmark again, due to (a) the overhead of the LLM judges, and (b) the narrow scope of what the benchmark seeks to measure. I would recommend reframing it as an empirical investigation of the physical intuitions of MLLMs, rather than a useful dataset for the practitioner.
7. MLLM and VLM seem to be used interchangeably throughout, starting in the abstract.
8. In Figure 1, the question for the 'Number' task is the same as the 'Size' task (Is the size of the playdough in the first image the same as in the final image?). Shouldn't it be something to do with coins (given the details in the list in Section 3.1).

### Missing Literature

There are some further studies on visual reasoning that are not mentioned:

1. Balazadeh, V., Ataei, M., Cheong, H., Khasahmadi, A. H., & Krishnan, R. G. (2025). Physics context builders: A modular framework for physical reasoning in vision-language models. In Proceedings of the IEEE/CVF International Conference on Computer Vision (pp. 7318-7328).

2. Campbell, D., Rane, S., Giallanza, T., De Sabbata, C. N., Ghods, K., Joshi, A., ... & Webb, T. (2024). Understanding the limits of vision language models through the lens of the binding problem. Advances in Neural Information Processing Systems, 37, 113436-113460.

3. Schulze Buschoff, L. M., Akata, E., Bethge, M., & Schulz, E. (2025). Visual cognition in multimodal large language models. Nature Machine Intelligence, 7(1), 96-106.

4. Schulze Buschoff, L. M., Voudouris, K., Akata, E., Bethge, M., Tenenbaum, J. B., & Schulz, E. (2025). Testing the limits of fine-tuning to improve reasoning in vision language models. arXiv preprint arXiv:2502.15678.

With respect to cognitive psychology, the line of work on the tunnel effect, object files, and object persistence should be discussed and mentioned:

1. Burke, L. 1952: On the tunnel effect. Quarterly Journal of Experimental Psychology, 4, 121 – 138.

2. Flombaum, J. I., & Scholl, B. J. (2006). A temporal same-object advantage in the tunnel effect: facilitated change detection for persisting objects. Journal of Experimental Psychology: Human Perception and Performance, 32(4), 840.

3. Flombaum, J. I., Kundey, S. M., Santos, L. R., & Scholl, B. J. (2004). Dynamic object individuation in rhesus macaques: A study of the tunnel effect. Psychological science, 15(12), 795-800.

4. Mitroff, S. R., Scholl, B. J., & Wynn, K. (2004). Divide and conquer: How object files adapt when a persisting object splits into two. Psychological Science, 15(6), 420-425.

5. Noles, N. S., Scholl, B. J., & Mitroff, S. R. (2005). The persistence of object file representations. Perception & Psychophysics, 67(2), 324-334.

6. Scholl, B. J. (2007). Object persistence in philosophy and psychology. Mind & Language, 22(5), 563-591.

**Questions:**

* This work and others all point to the conclusion that VLMs really aren't that good at 'intuitive physics'. And yet, they seem to perform really well on standard large-scale benchmarks and users love to use them. I'm intrigued whether the authors think that an inability to reason about conservation is actually *problematic* for VLM use and deployment?

---

> ### Author Response · Authors · 2025-11-26
> **rebuttal-weaknesses**
>
> ## Weaknesses
> > W1: The hybrid evaluation is an interesting solution to the problem of evaluating complex outputs, but using LLM judges incurs significant overhead for the practitioner. Since the paper currently relies only on open-source models (as far as I can tell) and the benchmark uses multiple choice questions, the authors could simply use the log-probability of the choice label, conditional on the text-image input. This could be normalised across the possible outcomes too. This is quite standard in benchmark evaluation (see EleutherAI's lm-evaluation-harness).
>
> > A1: Thanks for raising this question. While logit-based scoring is indeed a computational efficient approach. it is not compatible with reasoning models or VLMs that produce explicit reasoning traces before committing to an answer.
> To ensure a fair and consistent evaluation protocol, we thus adopt LLM-as-a-judge as our standard approach.
>
> ---
>
> > W3: Many of the tasks used here might be quite novel to the language model, so it's not clear whether these models could learn to reason about conservation if given the right training, or whether there is something deeper about architecture/pre-training that prevents them from doing so. Two further experiments are required to elucidate this. First, examining whether in-context learning can boost model performance. Here, examples of conservation/non-conservation with correct labels are given sequentially prior to the test question. Second, examining whether supervised fine-tuning on a subset of the tasks improves performance. There are three conditions of interest here: training on (a) a random subset of the tasks; (b) conservation problems; (c) non-conservation problems. I suspect, however, that fine-tuning would be just as brittle (see Schulze-Buschoff et al. 2025). This would make a more powerful point about VLMs. It's not just that current models off-the-shelf are incapable, but it's a feature of the architecture/large-scale pre-training that prevents them from having these common-sense intuitions.
>
> > A3: Thanks for the great suggestions! It indeed will make a more powerful point about VLMs on conservation capability. However, due to limited time and computation, we are not able to provide any experimental results at the moment. However, we will strive for the suggested training result towards the final camera-ready version of the paper.

---

> ### Author Response · Authors · 2025-11-26
> **rebuttal-weaknesses**
>
> > W4: I don't really think the white-image condition is a good control. To me, the bias towards invariance is shown by the delta between the conservation and non-conservation conditions. I would not expect systematic deviations from chance with white images, because where would that systematicity come from, absent some visual input? Perhaps I am missing the logic here.
>
> > A4: Thanks for raising this point. The white-image condition and the non-conservation condition are designed to probe **two different sources of bias**, and they serve complementary purposes.
> >
> > * **White-image condition:**
> >   Here the model receives *no visual evidence at all*. Any deviation from chance therefore, reflects **pure language-prior bias**—for example, a tendency to default to “conserve” whenever the question asks whether a quantity remains the same. This allows us to measure how strongly the model’s answer is driven by linguistic shortcuts rather than vision.
> >
> > * **Non-conservation condition:**
> >   In contrast, the model *does* receive visual input, but the correct answer reverses relative to the conservation case. If the model still answers “conserve,” this reflects **visual-insensitive bias**—i.e., treating conservation as the default regardless of what the images depict. This is a distinct failure mode: it shows that the model *has* visual input but does not integrate it appropriately.
> >
> > The logic is therefore not that white images should produce systematic patterns, but rather that they allow us to **separate language-only priors from visually induced heuristics**, making it possible to interpret the direction and magnitude of biases observed in the main conditions.
>
>
> ---
>
> > W5: There is some missing literature, which I include below.
>
> > A5 Thank you for the suggestions. We have incorporated all the suggested literature in Section 2.2.
>
> ---
>
> > W6: It's a semantic point, but I dispute that the main contribution of this paper is a benchmark. Rather, it's a careful series of experiments to test some hypotheses about MLLM capabilities. I don't think anyone will want to use this as a benchmark again, due to (a) the overhead of the LLM judges, and (b) the narrow scope of what the benchmark seeks to measure. I would recommend reframing it as an empirical investigation of the physical intuitions of MLLMs, rather than a useful dataset for the practitioner.
>
> > A6: Thanks for the thoughtful comment. Our intention is indeed to contribute **both**. We believe a great benchmark should do more than just provide data—it should measure critical aspects of the model and reveal significant insights. Our work, therefore, couples (1) a controlled series of experiments testing concrete hypotheses about VLM reasoning with (2) a carefully constructed dataset that future work can use to evaluate conservation reasoning under strictly controlled settings.
> >
> > We respectfully disagree with the claim that the benchmark is impractical or overly narrow:
> >
> > 1. **LLM judges are not a requirement.**
> >   The benchmark is formatted as multiple-choice VQA, consistent with many widely used MLLM benchmarks (e.g., SeedBench, MME). Practitioners who prefer a lightweight setup can use pattern matching or direct choice-label scoring. We use LLM judges only to ensure fair evaluation for reasoning models that produce long chains of thought and models that directly output an option, not because the benchmark intrinsically requires them.
> >
> > 2. **The scope is broader than “narrow perception.”**
> >   Conservation is not a low-level perceptual task. It requires integrating sequential visual evidence, maintaining transformation-stable internal representations, and distinguishing invariant from changing properties. These are fundamental components of physical reasoning. By revealing where models fail—especially under controlled manipulations—our benchmark provides a diagnostic testbed for probing physically grounded reasoning, not just another perceptual challenge set.
> >
> > In short, our empirical findings and the benchmark that produced them serve complementary purposes: the former characterizes current model limitations, and the latter offers a reusable, controlled environment for future research to evaluate and improve conservation reasoning in VLMs.
>
> ---
>
> > W7: MLLM and VLM seem to be used interchangeably throughout, starting in the abstract.
>
> > A7: Thank you for pointing this out. In the revised manuscript, we will use the term "VLM" throughout to ensure clarity.
>
> ---
>
> > W8: In Figure 1, the question for the 'Number' task is the same as the 'Size' task (Is the size of the playdough in the first image the same as in the final image?). Shouldn't it be something to do with coins (given the details in the list in Section 3.1).
>
> > A8: Thanks for pointing this out. We have corrected this in the revised manuscript.

---

> ### Author Response · Authors · 2025-11-26
> **rebuttal-questions**
>
> ## Questions
>
> > Q1: This work and others all point to the conclusion that VLMs really aren't that good at 'intuitive physics'. And yet, they seem to perform really well on standard large-scale benchmarks and users love to use them. I'm intrigued whether the authors think that an inability to reason about conservation is actually problematic for VLM use and deployment?
>
> > A1: Yes—we believe these failures matter, especially for applications where models must reason about physical transformations rather than recite patterns from training data. Conservation is a diagnostic case because it requires tracking how quantities and identities persist under change. A model that solves tasks through superficial heuristics can perform well on large-scale benchmarks yet fail unpredictably in real deployment, posing significant risks in physically grounded applications where reliable transformation reasoning is critical. This is a direct concern for embodied systems, robotics, autonomy, and any safety-critical setting where misunderstanding how objects transform can produce cascading errors.
> >
> > More broadly, conservation captures a foundational substrate of human physical reasoning that supports higher-level physical reasoning. Prior work [1, 2, 3] has argued that without such core representations, higher-level abilities remain brittle and fail to generalize beyond curated benchmarks, a hypothesis known as "core knowledge deficits.". Our findings fit this pattern: the models appear competent when tasks align with dataset statistics but fail once invariance must be inferred rather than observed. This suggests not just missing task exposure, but a deeper representational gap that limits reliability and generalizability in real-world scenarios when not grounded in core physical principles.
> >
> > [1] Lake, B.M., Ullman, T.D., Tenenbaum, J.B. and Gershman, S.J., 2017. Building machines that learn and think like people. Behavioral and brain sciences, 40, p.e253.
> >
> > [2] Li, Y., Gao, Q., Zhao, T., Wang, B., Sun, H., Lyu, H., Hawkins, R.D., Vasconcelos, N., Golan, T., Luo, D. and Deng, H., 2025. Core knowledge deficits in multi-modal language models. Proceedings of the Forty-second International Conference in Machine Learning. PMLR.
> >
> > [3] Cai, Z., Wang, Y., Sun, Q., Wang, R., Gu, C., Yin, W., Lin, Z., Yang, Z., Wei, C., Qian, O. and Pang, H.E., 2025. Holistic Evaluation of Multimodal LLMs on Spatial Intelligence. arXiv preprint arXiv:2508.13142.

---

> ### Comment · Reviewer_1A1m · 2025-11-27
>
> Thank you to the authors for their engagement with my feedback. I appreciate the argumentation, and I generally agree with them about logit-based probes and the control conditions. I disagree a bit about the downstream utility of the benchmark, but we can put that down to different research philosophies.
>
> Since my proposed experiments were not run (which would make the paper much more compelling) and I continue to believe that this paper offers incremental results to a relatively niche audience, I believe that my current score (6) is still appropriate and I do not intend to change it.

---

> ### Author Response · Authors · 2025-11-27
> **rebuttal to weaknesses 2**
>
> Thanks for the prompt response. We have now added our rebuttal for Weakness 2 below (delay due to the time required to complete the additional experiments):
>
> > W2: Many new models are also capable of processing videos. I would suspect that the next generation of VLM will process video data effectively. Therefore, the longevity of the benchmark would be ensured if video-based evaluation was a central contribution. The authors could conduct a small study with, say, Qwen-2.5-VL-7B (an open source video-language model) to see if there is any difference between performance on video and performance with frame sequences.
>
> > A2: Thank you for the thoughtful suggestion. We fully agree that video-based evaluation is critical. However, any video is inherently a sequence of frames, and VLMs process videos by ultimately operating on these frames—either all frames or a subset sampled at a predefined rate. In practice, nearly all contemporary VLMs process videos by uniformly sampling frames (e.g., 1 fps or a fixed number of frames) before performing inference.
> >
> > To directly address the reviewer’s concern, we conducted additional experiments using VLMs that explicitly advertise video-processing capabilities, evaluating them under their default video configurations (i.e., default frame sampling rate). We then compare their performance against our standard 16-frame protocol. The results are presented in the table below.
> >
> > | Model | Input Type | FPS / #Frames | Avg Acc | Strict Pairwise | Conservation | Non-Conserving |
> > |-------|------------|---- |---------|-----------------|--------------|----------------|
> > | Qwen3-VL-8B | Video | 2 fps | 0.4501 | 0.1269 | 0.4316 | 0.4687 |
> > | | Frame | 16 frames | 0.4097 | 0.0938 | 0.4931 | 0.3264 |
> > | InternVL3-8B | Video | 32 frames | 0.4227 | 0.0920 | 0.5764 | 0.2691 |
> > | | Frame | 16 frames | 0.4384 | 0.0695 | 0.6875 | 0.1892 |
> > | InternVL3.5-8B | Video | 32 frames | 0.4696 | 0.1493 | 0.5851 | 0.3542 |
> > | | Frame | 16 frames | 0.4931 | 0.1111 | 0.6510 | 0.3351 |
> > | Qwen2.5-VL-7B-Inst. | Video | 2 fps | 0.3672 | 0.0573 | 0.6372 | 0.0972 |
> > | | Frame | 16 frames | 0.4306 | 0.0486 | 0.6493 | 0.2118 |
> > | LLaVA-OneVision-Q2-7B | Video | 8 frames | 0.3871 | 0.0104 | 0.4844 | 0.2899 |
> > | | Frame | 16 frames | 0.3811 | 0.0208 | 0.4601 | 0.3021 |
> >
> > Overall, we found no consistent performance advantage for video over our 16-frame input. Across models, the differences in average accuracy remain minimal and appear to stem primarily from variations in the total number of frames processed due to varied frame-sampling rates across models (ranging from 8 to 32 frames for video inputs versus our standardized 16 frames). Such differences bear little significance compared to the characteristic failure patterns that persist across both video and 16-frame evaluations: models show near-chance overall performance, low strict pairwise accuracy, and systematic low non-conserving performance (compared to conservation). Crucially, the same characteristic failure modes persisted under both evaluation regimes: near-chance overall accuracy, low strict pairwise performance, and systematically poor accuracy on non-conserving controls relative to conserving trials.
> >
> > These consistent patterns indicate that the core limitations we identify—heavy reliance on superficial visual heuristics and an inability to maintain transformation-invariant representations—can not be mitigated by simply by providing more frames and alternation of input types. This reinforces the relevance of our benchmark for emerging video-capable models.
>
> Can you kindly highlight the proposed experiments that were not run so that we can strive our best to resolve them?

---

> > ### Comment · Reviewer_1A1m · 2025-11-27
> >
> > Thank you for these further results, and apologies for my lack of clarity. These video-based results are certainly interesting, but I was rather referring to the in-context-learning and fine-tuning experiments in my original point 3. It would be interesting to see if models can learn to do these tasks with some in-context labelled examples or with some in-weight learning through fine-tuning - if they can, it suggests they just need some calibration to the particular task they are setting them. Often, in psychology experiments, you allow the participant to gain some familiarity with the testing material so they can situate themselves in the task before testing begins. Otherwise there can be many reasons why they fail - they may misunderstand parts of the task or be attending to the wrong components of it. In-context or in-weight learning would mirror that. Subsequently, it would be really interesting to see if, for instance, fine-tuning confers some generalizable advantage to new conservation tasks with different numbers of objects or different visual features, but this would be a nice-to-have extension if time allowed it. This is what my original point 3 was suggesting.
> >
> > I realise that time is limited to complete such experiments, but I think I would need at least something like an in-context and fine-tuning experiment to further increase my score.

---

> > > ### Author Response · Authors · 2025-11-27
> > > **Response to reviewer**
> > >
> > > Thank you so much for the prompt response and clarification. We are working on in-context learning and fine-tuning right now to meet with your suggestions!
> > >
> > > Thanks,
> > > Author team

---

> > > ### Author Response · Authors · 2025-12-04
> > > **Further Response to Reviewer 1A1m: fine-tuning experiment**
> > >
> > > Many thanks again for the suggestions on further experiments. To address this, we fine-tuned a Qwen3-VL-8B model using LoRA on video data across both conservation tasks and non-conserving controls. Given the similarity between tasks within a single domain, models could potentially exploit spurious visual correlations specific to particular object or task types. To assess genuine conservation understanding versus pattern matching, we therefore employed a cross-validation approach: we held out one quantitative domain (Number, Liquid, Solid, or Length) and fine-tuned the model on the remaining three domains. We then evaluated performance on both a sample from the training domains previously reserved for evaluation (in-distribution, ID) and the complete video set for the held-out domain (out-of-distribution, OOD). The hypothesis is that if models improve not only on ID tasks but also generalize to the OOD domain—even with smaller gains—this would demonstrate within-construct transfer, suggesting models can acquire systematic conservation reasoning rather than memorizing task-specific heuristics. The results are as follows:
> > >
> > > | Hold-out Domain | Test Type | Base Model | LoRA Model | Improvement |
> > > |-----------------|-----------|------------|------------|-------------|
> > > | Number | ID | 26.67% | 55.56% | +28.89% |
> > > | | OOD | 46.88% | 46.88% | 0.00% |
> > > | Liquid | ID | 28.89% | 60.00% | +31.11% |
> > > | | OOD | 26.04% | 16.67% | -9.37% |
> > > | Solid | ID | 37.78% | 55.56% | +17.78% |
> > > | | OOD | 14.58% | 2.08% | -12.50% |
> > > | Length | ID | 31.11% | 53.33% | +22.22% |
> > > | | OOD | 50.00% | 46.88% | -3.12% |
> > >
> > > Fine-tuning yields substantial improvements on in-distribution tasks (+17.78% to +31.11%). However, performance on held-out domains shows no transfer or even degradation, with OOD improvements ranging from 0.00% to -12.50%. This suggests that fine-tuned models are acquiring domain-specific visual heuristics rather than conservation reasoning that generalizes across quantitative properties. The lack of within-construct transfer indicates that current VLMs struggle to extract abstract transformation-invariant principles from training data, instead overfitting to superficial features of the training domains. These findings further underscore the fundamental limitations in current models' capacity for systematic physical reasoning and highlight the diagnostic value of our benchmark.
> > >
> > > Due to time constraints, we are still implementing the in-context learning experiment and will include them in the camera-ready version of the manuscript. Nevertheless, we believe the fine-tuning experiment, in connection with the series of experiments already reported, are conclusive in demonstrating that current VLMs lack the capacity for systematic transformation reasoning.

---

### Official Review · Reviewer_z9U6 · 2025-10-28

**Soundness:** 2
**Presentation:** 2
**Contribution:** 2
**Rating:** 2
**Confidence:** 4

**Summary:**

The paper introduces ConservationBench, a video-based benchmark to test whether VLMs can reason about physical transformations by judging conservation (vs. matched non-conserving counterfactuals) of four quantities—number, length, volume, size. The suite varies temporal sampling (3/8/16 frames), frame-selection strategies (uniform, model-based, human-picked), and prompt types, producing 13824 trials. Across 34 models, the authors report performance only marginally above chance on average; improvements on conserving items often invert on matched non-conserving cases, suggesting brittle heuristics rather than true transformation reasoning. Human accuracy (on a subset) is ~95%. The paper concludes that current VLMs cannot reason about physical transformation.

**Strengths:**

- This paper focuses on conservation under transformation with a counterfactual non-conserving item.

- This paper provides the results under different prompt styles, frame counts, and frame-selection methods (uniform / human / SEVILA-style)

**Weaknesses:**

- The current evaluation setup only provides models maximum 16 frames. It is questionable that is this enough even for human to understand the physical transformation happening in the video. Therefore, the claim like “VLMs cannot reason about physical transformation” are overstated if the inputs to the models does not contain enough information to solve the task.

- The human baseline details are missing. How did you evaluate the human performance exactly?

- The paper does not evaluate state-of-the-art closed-source VLMs such as Openai, Claude, and Gemini models.

**Questions:**

- The question in Figure 1 for number case seems wrong.

- Please fix VLM/MLLM naming consistency (in abstract) and remove duplicated paragraphs (line 180-186).

- It would be helpful to include a full input prompt to the model including both image inputs and text inputs.

- What is the performance of the state-of-the-art closed-source VLMs?

---

> ### Author Response · Authors · 2025-11-26
> **rebuttal-weaknesses**
>
> > W1: The current evaluation setup only provides models maximum 16 frames. It is questionable that is this enough even for human to understand the physical transformation happening in the video. Therefore, the claim like “VLMs cannot reason about physical transformation” are overstated if the inputs to the models does not contain enough information to solve the task.
>
> > A1: Thanks for the concern. While 16 frames might be insufficient for long videos, our benchmark does not suffer from this issue for three reasons: 1. Most videos in our dataset are approximately 10 seconds long. With 16 frames, the model effectively receives more than 1 fps; 2. To ensure sufficient information, we further explore with different key frame selection methods; 3. Over decades of developmental research show that once key milestones are reached, humans from late childhood solve conservation tasks with robust, near-perfect accuracy, indicating the task presents little difficulty for human-level reasoning. In fact, our human participants on averaged scores a 95.25% accuracy on our benchmark, indicating that the stimuli contain sufficient information for successful inference, as shown in Section 4.2.
>
> ---
>
> > W2: The human baseline details are missing. How did you evaluate the human performance exactly?
>
> > A2: We evaluated human performance using the exact same stimuli and evaluation protocol as the VLMs (with the exception that participants directly selected answers rather than requiring LLM judges). Three adult participants (college students fluent in English) each completed one-third of a randomly sampled subset. They viewed the same frame sequences and answered the same three-choice questions presented to the models. The details of these procedural are now described more explicitly in the revised manuscript.
>
> ---
>
> > W3: The paper does not evaluate state-of-the-art closed-source VLMs such as Openai, Claude, and Gemini models.
>
> > A3: Thank you for raising this important point. To address this concern, we evaluated five leading closed-source VLMs: GPT-5 (OpenAI), Claude Sonnet 4.5 (Anthropic), Gemini 2.5 Pro (Google), Doubao Vision, and Hunyuan Vision.
> >
> > While these proprietary systems achieve higher overall accuracy than all open-source models, they exhibit the same fundamental failure modes: inverse performance patterns between conservation and non-conserving control tasks, and low strict pairwise accuracy, indicating reliance on default heuristics rather than genuine transformation reasoning.
> >
> > Here is a figure comparing the top 5 proprietary and top-5 open-source models (https://imgur.com/a/proprietary-vs-opensource-HE0O1bP). The table below also shows the performance for the top 10 models (5 proprietary and top-5 open-source models ranked by average accuracy):
> >
> > | Rank | Model | Type | Average Accuracy | Strict Pairwise | Conservation Task | Non-Conservation Task |
> > |------|-------|------|------------------|-------------------|-------------------|----------------------|
> > | 1 | Gemini 2.5 Pro | Proprietary | 0.6911 | 0.4120 | 0.9433 | 0.4388 |
> > | 2 | Doubao Vision | Proprietary | 0.6471 | 0.3822 | 0.8299 | 0.4642 |
> > | 3 | GPT-5 | Proprietary | 0.6365 | 0.3314 | 0.9148 | 0.3582 |
> > | 4 | Claude Sonnet 4.5 | Proprietary | 0.6041 | 0.3723 | 0.7543 | 0.4539 |
> > | 5 | Qwen3-VL-32B | Open-Source | 0.5831 | 0.3115 | 0.6457 | 0.5206 |
> > | 6 | Qwen2.5-VL-72B | Open-Source | 0.5583 | 0.1732 | 0.8969 | 0.2198 |
> > | 7 | Hunyuan Vision | Proprietary | 0.5505 | 0.2680 | 0.7405 | 0.3605 |
> > | 8 | InternVL3-38B | Open-Source | 0.5383 | 0.2974 | 0.6129 | 0.4636 |
> > | 9 | InternVL3-5-38B | Open-Source | 0.5364 | 0.1932 | 0.7549 | 0.3180 |
> > | 10 | InternVL3-5-38B-MPO | Open-Source | 0.5336 | 0.2085 | 0.7275 | 0.3398 |
> >
> > Despite higher overall accuracy, even the strongest proprietary models (e.g., Gemini 2.5 Pro, GPT-5) show severe asymmetries between conservation and non-conservation tasks (e.g., 94% vs. 44% for Gemini), and strict pairwise accuracy remains below 42% across all models. These patterns are consistent with the central claim of the paper: current VLMs—open-source and proprietary alike—lack robust reasoning about physical transformations.
> >
> > In the revised manuscript, we will add the full performance table and figure as well as the corresponding qualatative results of these models.

---

> ### Author Response · Authors · 2025-11-26
> **rebuttal-questions**
>
> > Q1: The question in Figure 1 for number case seems wrong.
>
> > A1: Thanks for pointing this out. We have corrected this in the revised manuscript.
>
> ---
>
> > Q2: Please fix VLM/MLLM naming consistency (in abstract) and remove duplicated paragraphs (line 180-186).
>
> > Q2: Thank you for noting these issues, which we have fixed in the revised manuscript.
>
> ---
>
> > Q3: It would be helpful to include a full input prompt to the model including both image inputs and text inputs.
>
> > A3: Thanks for the suggestion. We have added a full example of the model input prompt in the appendix, including both the image inputs and the corresponding text query.
>
> ---
>
> > Q4: What is the performance of the state-of-the-art closed-source VLMs?
>
> > A4: please refer to W3.

---

### Official Review · Reviewer_X9Du · 2025-10-31

**Soundness:** 3
**Presentation:** 2
**Contribution:** 2
**Rating:** 4
**Confidence:** 3

**Summary:**

The paper introduces a benchmark to test if models understand that some physical properties remain invariant under transformations. They show that models can not keep track of conserved properties very well. While testing whether models can keep track of physical properties is a nice idea, I feel the scope of the paper and the insights it delivers are too limited to recommend acceptance.

**Strengths:**

**Originality** I'm not aware of any previous work investigating conservation in vision language models.

**Quality** The benchmark is well constructed.

**Clarity** The paper follows a clear outline, although I feel the writing is a bit bloated and unaccessible at parts.

**Significance** The paper adds to a growing body of work highlighting the shortcomings of vision language models with basic visual processing. I like the split of the benchmark in conserving vs. non-conserving stimuli. Also, the result that models seem to default to either, always saying a property is conserved or always saying it has changed, regardless of the actual transformation, is interesting. However, there have been a large number of papers evaluating specific visual properties in vision language models with specific datasets/benchmarks.

**Weaknesses:**

In general, I feel like this paper does not provide a strong enough novel contribution to recommend acceptance. There is at this point a large growing body of evidence that vision language models fail at very basic visual processing. While this paper adds some novel data to this pile of findings, I find that the aspect of perception that it investigates is just too narrow. Also, the authors do not offer concrete ideas on how these problems could be overcome.

**Questions:**

**Main questions**
- How is the chance rate that is mentioned throughout at 33% if the models are always asked a binary question (does the property change versus does it not change)? Is this because you map the answer either to one of the two options or to "Fail" if it can't be parsed correctly? This is not the most obvious way of computing a baseline for me, if we accept that the models could output anything and LLM judges finally decide if the output maps to one of the two answer options, the chance level is not obvious to calculate. In any case, I think the reasoning for why you set it to 33.3% should be made more transparent in the text.
- It's interesting and a bit strange to me that the CoT prompting performs the worst. Could you maybe speculate a bit on why that is?
- For the human evaluation, you write "The aggregated human accuracy reaches 95.25%". Just to make sure, this is the accuracy given videos with all frames and in the "non-strict" evaluation, right? I think this could be made a bit more clear.
- Initially I thought the number of frames (3, 8, 16) would be combined with the different methods of selection (Uniform, Human-selected, SeViLA), but here you seem to report them separately in Figure 3. For clarity, B shows uniformly sampled 3, 8, and 16 frames, right? And for C, is the number of frames fixed for all three selection methods and if so, what is it? Again, I may have missed these details in the text but feel they could be outlined more clearly, maybe even in the caption of the Figure.

**Minor comments**
- I think the abstract is too long and should be cut down.
- Line 43 "Yet it remains unclear whether VLMs possess a true understanding of physical principles or the capacity to operate reliably in embodied physical environments." there is previous work that shows VLMs do not understand physical principles [1, 2].
- Line 82 comes out of nowhere "Physical quantity refers to the measurable magnitude of objects along certain dimensions, while spatial transformation denotes the continuous processes through which objects change in appearance under perception." What is this in reference to? It seems a bit misplaced.
- Line 315 " 95. 25%", there's likely a space too much.
- Figure 2A the colors in the legends do not match the color of the bars in the plot?
- Figure 2 captions and titles are a not coherent, "Non-conserving" in the caption and "Non-conserve" in the title.

[1] Schulze Buschoff, Luca M., et al. "Visual cognition in multimodal large language models." Nature Machine Intelligence 7.1 (2025): 96-106.

[2] Balazadeh, Vahid, et al. "Synthetic vision: Training vision-language models to understand physics." arXiv e-prints (2024): arXiv-2412.

---

> ### Author Response · Authors · 2025-11-26
> **rebuttal-weaknesses and main questions(1/2)**
>
> ## Weaknesses
>
> > W: In general, I feel like this paper does not provide a strong enough novel contribution to recommend acceptance. There is at this point a large growing body of evidence that vision language models fail at very basic visual processing. While this paper adds some novel data to this pile of findings, I find that the aspect of perception that it investigates is just too narrow. Also, the authors do not offer concrete ideas on how these problems could be overcome.
>
> > A: We respectfully disagree with the characterization that conservation tasks probe only “basic visual processing.” As we discuss in the manuscript, conservation relies on perceptual inputs but is fundamentally a reasoning problem: the observer must integrate sequential visual evidence, maintain stable internal representations across transformations, and distinguish invariant from changing properties. These requirements go well beyond low-level perception such as boundary detection or size estimation.
> >
> > Moreover, our benchmark is not another dataset documenting failures on simple visual primitives. It targets a core component of physical reasoning that existing evaluations largely ignore—tracking identity and quantity through transformations. The failure modes we uncover (e.g., reliance on default heuristics that systematically flip in non-conserving controls, and indifference to increased temporal resolution) point to structural weaknesses in models’ causal and physical reasoning, not deficiencies in visual processing.
> >
> > While proposing concrete architectural remedies is outside the scope of this empirical study, our findings identify where models break and why current training regimes do not yield stable representations of physical invariance. We see this as a necessary precursor to developing targeted solutions.
>
> ---
>
> ## Main questions
>
> > Q1: How is the chance rate that is mentioned throughout at 33% if the models are always asked a binary question (does the property change versus does it not change)? Is this because you map the answer either to one of the two options or to "Fail" if it can't be parsed correctly? This is not the most obvious way of computing a baseline for me, if we accept that the models could output anything and LLM judges finally decide if the output maps to one of the two answer options, the chance level is not obvious to calculate. In any case, I think the reasoning for why you set it to 33.3% should be made more transparent in the text.
>
> > A1: Thanks for raising this question. To clarify, the chance rate is 33.3\% because, in all experiments, models are explicitly required to choose among three mutually exclusive options. For instance, in the number-conservation tasks, the response set includes: (i) “No, the lower row has more coins,” (ii) “No, the upper row has more coins,” and (iii) “Yes, they are the same.” The LLM judge subsequently maps each model output to one of these three predefined options, or labels it as Fail when it cannot be parsed into any option. Thus, under random responding, the probability of selecting the correct option is 1/3. We have included example full prompts in the revised manuscript and highlighted the three-choice structure more explicitly to avoid confusion about the baseline calculation.
>
> ---
>
> > Q2: It's interesting and a bit strange to me that the CoT prompting performs the worst. Could you maybe speculate a bit on why that is?
>
> > A2: Thanks for raising this comment. We believe this phenomenon stems from a mismatch between what conservation tasks demand (the intuitive nature of conservation reasoning) and what CoT prompting actually elicits. Conservation reasoning, while involving specific mental operations, is fundamentally intuitive in humans—driven by stable internal representations rather than slow, step-by-step verbal reasoning. CoT, by contrast, forces models into an explicit “System 2–style” narrative that they are not equipped to ground in correct physical understanding.
> >
> > Because current VLMs lack genuine representations of physical transformation, CoT tends to amplify incorrect heuristics. When prompted to explain their answers, models often verbalize shallow cues (e.g., “the row looks longer, so there must be more objects”), locking themselves into erroneous reasoning paths that would not necessarily arise in direct-answer mode. In this sense, CoT acts as a confabulation amplifier: the model produces fluent, coherent explanations that rationalize faulty perceptual intuitions rather than correcting them.

---

> ### Author Response · Authors · 2025-11-26
> **rebuttal-main questions (2/2)**
>
> ## Main questions
>
> > Q3: For the human evaluation, you write "The aggregated human accuracy reaches 95.25%". Just to make sure, this is the accuracy given videos with all frames and in the "non-strict" evaluation, right? I think this could be made a bit clearer.
>
> > A3: Yes, that is correct. We have clarified this in the revised manuscript to avoid ambiguities.
>
> ---
>
>
> > Q4: Initially I thought the number of frames (3, 8, 16) would be combined with the different methods of selection (Uniform, Human-selected, SeViLA), but here you seem to report them separately in Figure 3. For clarity, B shows uniformly sampled 3, 8, and 16 frames, right? And for C, is the number of frames fixed for all three selection methods and if so, what is it? Again, I may have missed these details in the text but feel they could be outlined more clearly, maybe even in the caption of the Figure.
>
> > A4: Thank you for pointing this out. In our experimental setup, prompts (Direct, Sequential, CoT, Continuous), frame count (3, 8, 16), and extraction method (Uniform, Human-selected, SeViLA) form factorial combinations, yielding 36 conditions total. In Figure 3, we report **main** effects rather than all 36 cells to maintain interpretability:
> >
> > * **Panel A** shows accuracy for each prompt type averaged across all frame counts and extraction methods.
> > * **Panel B** shows accuracy for each frame count averaged across all prompts and extraction methods.
> > * **Panel C** shows accuracy for each extraction method averaged across all prompts and frame counts.
> >
> > This presentation isolates the contribution of each factor and keeps the figure interpretable. We agree that this averaging could be stated more explicitly. In the revised manuscript, we have updated the Figure 3 caption and added a brief note in Results clarifying that the full factorial design is used, and main-effect plots are produced by averaging over the complementary dimension.

---

> ### Author Response · Authors · 2025-11-26
> **rebuttal-minor comments**
>
> ## Minor Comments:
>
> > Q5: I think the abstract is too long and should be cut down.
>
> > A5: Thank you for the suggestion. We have revised and shortened the abstract for improved conciseness.
>
> ---
>
> > Q6: Line 43 "Yet it remains unclear whether VLMs possess a true understanding of physical principles or the capacity to operate reliably in embodied physical environments." there is previous work that shows VLMs do not understand physical principles [1, 2].
>
> > A6: Thanks for pointing this out. We have added these literatures in our related works section.
>
> ---
>
> > Q7: Line 82 comes out of nowhere "Physical quantity refers to the measurable magnitude of objects along certain dimensions, while spatial transformation denotes the continuous processes through which objects change in appearance under perception." What is this in reference to? It seems a bit misplaced.
>
> > A7: Thank you for this feedback. This sentence was intended to clarify the technical terms used in our definition of conservation, but we agree it disrupted the flow. We have adjusted the writing to integrate these definitions more smoothly.
>
> ---
>
> > Q8: Line 315 " 95. 25%", there's likely a space too much. Figure 2A the colors in the legends do not match the color of the bars in the plot? Figure 2 captions and titles are a not coherent, "Non-conserving" in the caption and "Non-conserve" in the title.
>
> > A8: Thanks for pointing them out. We have fixed it in the revised manuscript.

---

> > ### Comment · Reviewer_X9Du · 2025-11-27
> > **Reviewer response to rebuttal**
> >
> > Dear Authors,
> >
> > I appreciate the detailed responses to my concerns, thank you. I want to remind the authors to update the manuscript in open review.
> >
> > While my more specific methodological questions were answered, I do not think the authors make a sufficiently good case for the contribution of this work. While there may be some debate on what exactly constitutes "basic visual processing", this work much more than being a benchmark (as also pointed out by reviewer 1A1m), is to me an investigation of specific aspects of vision language models' visual processing and perhaps reasoning) abilites. While this paper introduces novel insights into models' ability to keep track of phyiscal properties, there exist a number of works focusing on other specific visual processing capabilities and I feel at this point adding more evidence in the form of empirical studies does not constitute a significant enough contribution.
> >
> > To put it bluntly, before reading this paper I knew that vision language models were not good at reasoning about physical processes. After reading the paper I know more about specific failure models when it comes to conservation of physical properties, but I still do not know why they struggle with this, or how these problems could be overcome. I feel this is a quite small incremental gain in knowledge, and since you do not offer solutions to improve models' abilities, or test specific architectures that should be better equipped for solving these types of problems, I can not recommend acceptance at this point.

---

> > > ### Author Response · Authors · 2025-11-29
> > > **Response to Reviewer X9Du (part 1)**
> > >
> > > Dear Reviewer X9Du,
> > >
> > > Thank you for your time and efforts in the reply. We appreciate your concerns and shares your interests in further this line of work. However, we wish to make a few clarifications:
> > >
> > > > While there may be some debate on what exactly constitutes "basic visual processing", this work much more than being a benchmark (as also pointed out by reviewer 1A1m), is to me an investigation of specific aspects of vision language models' visual processing and perhaps reasoning) abilites.
> > >
> > > We wish to highlight that decades of cognitive scientific research have established that tasks like conservation, which require the maintenance of structured internal representations across transformations, are well beyond "basic visual processing" [1,2,3]. This distinguishes our study from other benchmarks focusing on visual reasoning that can be resolved from static images.
> > >
> > > > While this paper introduces novel insights into models' ability to keep track of phyiscal properties, there exist a number of works focusing on other specific visual processing capabilities and I feel at this point adding more evidence in the form of empirical studies does not constitute a significant enough contribution.
> > >
> > > While we agree that our study provides a deep-dive investigation into this phenomenon, that does not undermine its value as a benchmark. In particular, we believe the tasks curated in this study can serve as *sanity checks* for foundation models' transformation reasoning as the field progresses in high-level reasoning while robustness challenges prevail. This speaks directly to the concern of "core knowledge deficits" established in previous work [4,5]. Our new results described in the general comment—showcasing the inverse trend between conservation and non-conserving control performance across 109 VLMs released between January 2024 and October 2025—further demonstrate the benchmark's value as an enduring diagnostic test.
> > >
> > > > To put it bluntly, before reading this paper I knew that vision language models were not good at reasoning about physical processes. After reading the paper I know more about specific failure models when it comes to conservation of physical properties, but I still do not know why they struggle with this, or how these problems could be overcome.
> > >
> > > We appreciate this perspective. However, we wish to highlight that our study establishes a rigorous test of transformation reasoning largely absent in previous empirical literature, and validating such concerns is precisely the value of our work. The recent breakthroughs of VLMs on physical reasoning benchmarks make it increasingly important to validate whether these capabilities reflect genuine understanding or superficial pattern matching.
> > >
> > > We share the interest in understanding the deeper reasons behind these failures and have several hypotheses under investigation. It remains a possibility that surface-level interventions regarding the targeted concept and scenarios may help, and we are actively implementing ICL and fine-tuning experiments using curated knowledge supplementation and labeled data (as suggested by Reviewer 1A1M). If such targeted interventions prove insufficient, this would support recent hypotheses that VLMs are architecturally limited in structured spatial/physical reasoning due to, for example, coarse-grained visual encodings [6,7]. We speculate this is likely given the distinctive failure modes uncovered by our benchmark—particularly the systematic reversal of heuristics and insensitivity to temporal resolution. Our findings thus provide empirical evidence for such hypotheses emerging in recent literature, offering diagnostic clarity that can guide future architectural and training improvements.

---

> > > ### Author Response · Authors · 2025-11-29
> > > **Response to Reviewer X9Du (part 2)**
> > >
> > > > I feel this is a quite small incremental gain in knowledge, and since you do not offer solutions to improve models' abilities, or test specific architectures that should be better equipped for solving these types of problems, I can not recommend acceptance at this point.
> > >
> > > To this end, we wish to argue that establishing rigorous empirical evidence of systematic failures in transformation reasoning constitutes a necessary foundation for developing solutions. We have actively implemented a series of experiments to further investigate these phenomena, as per reviewer suggestions. However, we would also like to highlight that since this work focuses on VLMs—which remain a substantial sector of research and deployment interest—comprehensively testing alternative architectures or proposing entirely new model designs is beyond our current scope. Our contribution lies in providing diagnostic clarity about where and how current models fail, which is essential for guiding future innovations. We believe this does not constitute a weakness of the study itself, given the series of contributions we highlighted above: a cognitively-grounded benchmark, diagnostic methodology distinguishing heuristics from genuine reasoning, and systematic evidence of failure modes persisting across models. Thus, we would respectfully disagree with this characterization and hope you may reconsider in light of the benchmark's diagnostic value and implications for future research.
> > >
> > >
> > > [1] Piaget, J., & Inhelder, B. (2014). Intellectual operations and their development. In Experimental Psychology Its Scope and Method: Volume VII (Psychology Revivals) (pp. 144-205). Psychology Press.
> > >
> > > [2] Johnson-Laird, P. N. (1983). Mental models: Towards a cognitive science of language, inference, and consciousness (No. 6). Harvard University Press.
> > >
> > > [3] Houdé, O. (1997). Numerical development: From the infant to the child. Cognitive Development, 12(3), 373-391.
> > >
> > > [4] Li, Y., Gao, Q., Zhao, T., Wang, B., Sun, H., Lyu, H., Hawkins, R.D., Vasconcelos, N., Golan, T., Luo, D. and Deng, H., 2025. Core knowledge deficits in multi-modal language models. Proceedings of the Forty-second International Conference in Machine Learning. PMLR.
> > >
> > > [5] Cai, Z., Wang, Y., Sun, Q., Wang, R., Gu, C., Yin, W., Lin, Z., Yang, Z., Wei, C., Qian, O. and Pang, H.E., 2025. Holistic Evaluation of Multimodal LLMs on Spatial Intelligence. arXiv preprint arXiv:2508.13142.
> > >
> > > [6] Zhang, J., Hu, J., Khayatkhoei, M., Ilievski, F., & Sun, M. (2024). Exploring perceptual limitation of multimodal large language models. arXiv preprint arXiv:2402.07384.
> > >
> > > [7] Luo, D., Gao, Q., & Deng, H. (2025). Rethinking the Simulation vs. Rendering Dichotomy: No Free Lunch in Spatial World Modelling. arXiv preprint arXiv:2510.20835.

---

> > > ### Author Response · Authors · 2025-12-04
> > > **Response to Reviewer X9Du (follow-up on fine-tuning experiment)**
> > >
> > > Thanks again for your valuable feedback! We have now completed the fine-tuning experiments mentioned above. Using a leave-one-domain-out approach with LoRA fine-tuning on Qwen3-VL-8B, we observed substantial improvements on in-distribution tasks but no improvement—or even performance degradation—on held-out domains (please see full details in our latest response to Reviewer 1A1M).  This confirms that targeted interventions through supervised fine-tuning are still insufficient for acquiring conservation reasoning. Models appear to overfit to task-specific visual heuristics rather than extracting domain-general transformation-invariant principles. These results support the hypothesis that current VLMs face fundamental limitations in systematic physical reasoning, consistent with the diagnostic patterns documented throughout our benchmark.
> > >
> > > Again, we maintain that establishing such diagnostic clarity about where and how models fail—and demonstrating that surface-level interventions are insufficient—constitutes a necessary foundation for developing genuine solutions. This empirical evidence provides the kind of rigorous grounding that can guide future architectural innovations and targeted research directions.

---

### Author Response · Authors · 2025-11-27
**General Comment**

General Response to Reviewers

We sincerely thank all reviewers for their insightful and constructive feedback. Below we provide an overview of: (1) our general contributions, (2) changes made to the manuscript, and (3) additional experiments conducted in response to reviewer suggestions.

1. Key Contributions
Our work makes the following contributions:

* **Targets transformation reasoning beyond basic visual processing**: Drawing on developmental psychology, we specifically targets conservation as a key benchmark for transformation reasoning, which requires integrating sequential evidence, maintaining stable representations, and distinguishing invariant from changing properties, not just low-level perception.
* **Establishes diagnostic methodology**: Our paired conserving/non-conserving design with strict pairwise evaluation distinguishes superficial heuristics from genuine transformation understanding.
* **Identifies systematic reasoning failures**: We uncover specific failure modes (default heuristics that reverse in non-conserving controls, indifference to temporal resolution) revealing structural weaknesses in physical reasoning.
* **Provides cognitively-grounded evaluation with practical implications**: Our benchmark measures a foundational capacity that scaffolds higher-level physical reasoning in humans. The failures we document have direct implications for deployment in embodied systems, robotics, and safety-critical applications where reliable transformation reasoning is essential.

2. Documented Changes
In response to reviewer feedback, we have made the following revisions:

* Clarified human evaluation details.
* Clarified that panels show main effects averaged across other factors in the full factorial design
* Incorporated suggested references on VLM visual reasoning limitations and related cognitive scientific studies
* Copy-editing corrections: Fixed VLM/MLLM terminology consistency throughout; corrected Figure 1 number task question; removed duplicated paragraphs
* Shortened abstract: Revised for improved conciseness per reviewer suggestion

3. Additional Experiments

* To address reviewer concerns, we conducted three major additional evaluations. Beyond proprietary model evaluation (see detailed response A3 to Reviewer z9U6) and video-frame input comparison (see detailed response A2 to Reviewer 1A1m), we conducted substantial further experiments across 75 additional VLMs. Complete results will be reported comprehensively in further revisions of the paper.
* To this end, we analyzed performance trends across 109 VLMs released between January 2024 and October 2025, reported at this anonymous link (https://imgur.com/a/aAEsDkH). The results reveal only modest improvements in average accuracy over time. Specifically, while conservation task accuracy shows a slight upward trend, non-conserving control task accuracy exhibits a striking downward trend. This inverse relationship validates our core finding: models rely on default heuristics rather than genuine transformation reasoning. The persistent trade-off between conservation and non-conserving control performance, even as model scale and release date advance, demonstrates that current training paradigms fail to instill systematic physical understanding.

---

### Meta-Review · Area_Chair_h6fG · 2025-12-23

**Summary:**

This paper introduces a new benchmark testing VLM abilities for reasoning over conservation using clips. The paper demonstrates that current VLMs fail at this task.

The paper received mixed reviews with all three reviewers commenting on the limited significance of the findings. Several reviewers suggested additional analysis, including: 1) testing the effect of in-context learning & supervised fine-tuning to confirm whether VLMs fundamentally cannot reason about transformation or just whether VLMs have not yet been taught to reason; and 2) testing whether the results translate to state-of-the-art models.

Although I enjoyed the topic of the paper and I feel it has promise, I agree with concerns of limited significance of the benchmark and overstated claims about whether VLMs can or cannot reason about transformation. The rebuttal includes additional experiments that partially answer these questions. Given the additional work needed to incorporate these results and the remaining questions, I recommend a thorough revision and resubmission in the next cycle.

**Reviewer Concerns:**

Multiple reviewers had concerns about the limited significance of the findings, specifically, “why VLMs struggle with this [task], or how the problems could be overcome”. The rebuttal clarified the importance of the conservation framework and demonstrated numerical experiments showing finetuning is insufficient for acquiring conservation reasoning out-of-distribution. Reviewers recommended a set of meaningful experiments which may elucidate further insights. The rebuttal includes some of these experiments, but not all. Moreover, the scale of these experiments warrant proper inclusion into the paper. As is, the rebuttal does not give a conclusive guidance confirming that the issue is a fundamental architectural problem rather than something that could be fixed by some form of training/prompting.

**Reviewer Scores:**

The borderline positive reviewer and one negative reviewer specifically confirmed that their review would not change. The remaining negative reviewer held fundamental concerns about overstated claims. Given the common threads between all reviewers, I do not believe they would change scores.

---

### Decision · Program_Chairs · 2026-01-26

Reject